# Deep learning for the detection of benign and malignant pulmonary nodules in non-screening chest CT scans

Ward Hendrix [1,2], Nils Hendrix[1,2,3], Ernst T. Scholten[1], Mariëlle Mourits [4], Joline Trap-de Jong[5], Steven Schalekamp[1], Mike Korst[2], Maarten van Leuken[4], Bram van Ginneken[1], Mathias Prokop[1,6], Matthieu Rutten [1,2] & Colin Jacobs [1 ✉]

## Abstract

**Background** Outside a screening program, early-stage lung cancer is generally diagnosed after the detection of incidental nodules in clinically ordered chest CT scans. Despite the advances in artificial intelligence (AI) systems for lung cancer detection, clinical validation of these systems is lacking in a non-screening setting.

**Method** We developed a deep learning-based AI system and assessed its performance for the detection of actionable benign nodules (requiring follow-up), small lung cancers, and pulmonary metastases in CT scans acquired in two Dutch hospitals (internal and external validation). A panel of five thoracic radiologists labeled all nodules, and two additional radiologists verified the nodule malignancy status and searched for any missed cancers using data from the national Netherlands Cancer Registry. The detection performance was evaluated by measuring the sensitivity at predefined false positive rates on a free receiver operating characteristic curve and was compared with the panel of radiologists.

**Results** On the external test set (100 scans from 100 patients), the sensitivity of the AI system for detecting benign nodules, primary lung cancers, and metastases is respectively 94.3% (82/87, 95% CI: 88.1–98.8%), 96.9% (31/32, 95% CI: 91.7–100%), and 92.0% (104/113, 95% CI: 88.5–95.5%) at a clinically acceptable operating point of 1 false positive per scan (FP/s). These sensitivities are comparable to or higher than the radiologists, albeit with a slightly higher FP/s (average difference of 0.6).

**Conclusions** The AI system reliably detects benign and malignant pulmonary nodules in clinically indicated CT scans and can potentially assist radiologists in this setting.

**Plain language summary**

Early-stage lung cancer can be diagnosed after identifying an abnormal spot on a chest CT scan ordered for other medical reasons. These spots or lung nodules can be overlooked by radiologists, as they are not necessarily the focus of an examination and can be as small as a few millimeters. Software using Artificial Intelligence (AI) technology has proven to be successful for aiding radiologists in this task, but its performance is understudied outside a lung cancer screening setting. We therefore developed and validated AI software for the detection of cancerous nodules or non-cancerous nodules that would need attention. We show that the software can reliably detect these nodules in a non-screening setting and could potentially aid radiologists in daily clinical practice.

[1] Diagnostic Imaging Analysis Group, Radiology and Nuclear Medicine Department, Radboud University Medical Center, Geert Grooteplein Zuid 10, 6525 GA Nijmegen, The Netherlands. [2] Radiology Department, Jeroen Bosch Hospital, Henri Dunantstraat 1, 5223 GZ 's-Hertogenbosch, The Netherlands. [3] Jheronimus Academy of Data Science, Sint Janssingel 92, 5211 DA 's-Hertogenbosch, The Netherlands. [4] Radiology Department, Canisius Wilhelmina Hospital, Weg door Jonkerbos 100, 6532 SZ Nijmegen, The Netherlands. [5] Radiology Department, St. Antonius Hospital, Koekoekslaan 1, 3435 CM Nieuwegein, The Netherlands. [6] Radiology Department, University Medical Center Groningen, Hanzeplein 1, 9713 GZ Groningen, The Netherlands. ✉email: colin.jacobs@radboudumc.nl

Lung cancer is one of the most frequent cancers and is worldwide the leading cause of cancer death[1]. It has been estimated that 2.2 million global new lung cancer cases were diagnosed in 2020 and that 1.8 million cancer deaths were caused by lung cancer, almost one fifth of all cancer deaths. Lung cancer is often diagnosed at an advanced stage, because symptoms usually occur when the disease has progressed to a higher stage[2].

The survival rate substantially improves if lung cancer is diagnosed at an early stage[2]. For this reason, lung cancer screening programs aim to detect lung cancer as early as possible with low-dose computed tomography (CT), when the cancer presents as a small pulmonary nodule. Although trials have provided evidence that CT screening can substantially reduce lung cancer related mortality in a high-risk population[3,4], their implementation into clinical practice has been slow[5,6]. Hence, early-stage lung cancer is generally diagnosed after the detection of incidental nodules in non-screening chest CT scans that were ordered for other medical reasons[7]. Similarly, pulmonary metastases from extra-thoracic malignancies can be detected as incidental nodules as well and should be diagnosed and treated as early as possible given their large potential for further tumor spread[8].

However, the detection of pulmonary nodules in CT scans is a challenging task in a routine clinical setting. First, nodules can be as small as three millimeters and are therefore hard to detect by radiologists[9]. This is especially true when the diagnosis of lung and airway diseases is not the focus of the examination and the chosen imaging parameters are suboptimal for this task. Second, radiologists may only focus on the main clinical question and discontinue their search for additional findings such as nodules[10,11]. Finally, the workload of radiologists has dramatically increased in the past 15 years, mainly caused by the increasing number of CT studies[12]. This underlines the importance of efficient nodule detection and management.

Artificial intelligence (AI) is a potential solution to support radiologists with this task. Many AI studies have reported a high performance of deep learning-based computer-aided detection (DL-CAD) systems for nodule detection: the sensitivities range from 86% to 98% with an average of 1–2 false positives per scan (FP/s) on public CT datasets[13], such as the LIDC-IDRI dataset[14]. When using a DL-CAD system as a concurrent reader, radiologists can obtain a higher detection sensitivity, improve the uniformity of their management recommendations, and reduce their reading time[15–18].

Nonetheless, validation studies of DL-CAD systems on modern clinical datasets remain sparse[6,19]. Even when such a dataset is available, AI studies are often limited to a reference standard set by one or two radiologists while substantial interobserver variability exists for the task of nodule identification[20,21]. More importantly, most reference standards lack histopathological proof or follow-up imaging for at least 2 years to determine which individual nodules were malignant. Although multiple AI studies have already demonstrated the potential clinical value of DL-CAD systems for scan-level lung cancer detection[22–24], they do not assess their performance for detecting all clinically relevant nodules that require follow-up regardless of their malignancy status.

Therefore, the aim of this study is to bridge the gap between lung cancer and nodule detection AI studies. In a retrospective multi-center setting, we developed and validated a deep learning-based algorithm for the detection of pulmonary nodules in routine clinical CT scans with a reliable reference standard based on nodule identifications of five thoracic radiologists and nodule-level malignancy status. We demonstrate that a DL-CAD system can accurately detect benign pulmonary nodules, small lung cancers, and metastases in heterogenous CT scans that are made in routine clinical care.

## Methods

**Study design**. At both institutions, the local institutional review board approved the study and waived the need for informed consent because of the retrospective design and the use of anonymized data (Radboud University Medical Center: case 2016-3045, project 19010; Jeroen Bosch Hospital: case 2020.04.22.01). First, we developed and validated a pulmonary nodule detection system with the publicly available LUNA16 dataset[25], a subset of the LIDC-IDRI archive[14]. The details of this procedure and results are provided in Supplementary Note, Figure, and Table 1. Then, a large dataset of CT scans was collected from the picture archiving and communications systems (PACS) from a university medical center (hospital A; Radboud University Medical Center) and a large non-academic teaching hospital (hospital B; Jeroen Bosch Hospital) in the Netherlands. The CT scans from hospital A were annotated for the presence of pulmonary nodules by trained medical students and subsequently the system was re-trained using both the CT scans from LUNA16 and hospital A (see Supplementary Note 2 for more details). Finally, the detection system was evaluated on two datasets: a hold-out set with CT scans from hospital A (internal test set) and a completely independent set from hospital B (external test set). Five thoracic radiologists independently located the pulmonary nodules in the scans and two additional radiologists determined the malignancy status of each nodule and located any missed cancers using data from the national Netherlands Cancer Registry (NCR).

**Datasets**. For training a lung detection component of the AI system (see section Nodule detection pipeline), a dataset was prepared with 500 thorax and thorax-abdomen CT scans (500 patients) from hospital A from 2017. For training the nodule detection system, another dataset was prepared including all 888 thorax CT scans (887 patients) from the LUNA16 challenge[25] and 602 thorax CT scans (602 patients) from hospital A from 2017. For testing the complete nodule detection system, two datasets were prepared: one dataset with 100 thorax and thorax-abdomen CT scans (100 patients) from hospital A from 2018-2020 (internal test set) and another dataset including 100 thorax and thorax–abdomen CT scans from hospital B from the same period (external test set). At both hospitals, the scans were evenly sampled from four categories to obtain balanced datasets: (1) patients with stage I lung cancer; (2) patients with pulmonary metastases; (3) patients with benign pulmonary nodules larger than 5 mm for which imaging follow-up would be recommended[26]; and (4) patients with benign pulmonary nodules smaller than 5 mm or no nodules (which were considered as normal). There was no patient overlap between the training and test datasets.

Flowcharts of the study selection for the test and training datasets are shown in Fig. 1 and Supplementary Fig. 2, respectively. During the study selection procedure, an experienced radiologist (E.T.S) assessed the validity of the scans using the eligibility criteria as defined in the next section. In case of doubt, another radiologist was consulted (M.R.). The sample sizes of the training data obtained at hospital A were based on the size of the LUNA16 dataset. For the test set, we aimed to collect 100 scans per hospital, which was mainly determined by the maximum number of scans that could be annotated by the panel. One CT scan per patient was sampled to maximize the diversity of the datasets.

An overview of the main characteristics of each dataset is included Table 1 and additional imaging parameters are included in Supplementary Table 2. The annotation protocols for the training data are described in detail in Supplementary Note 3.

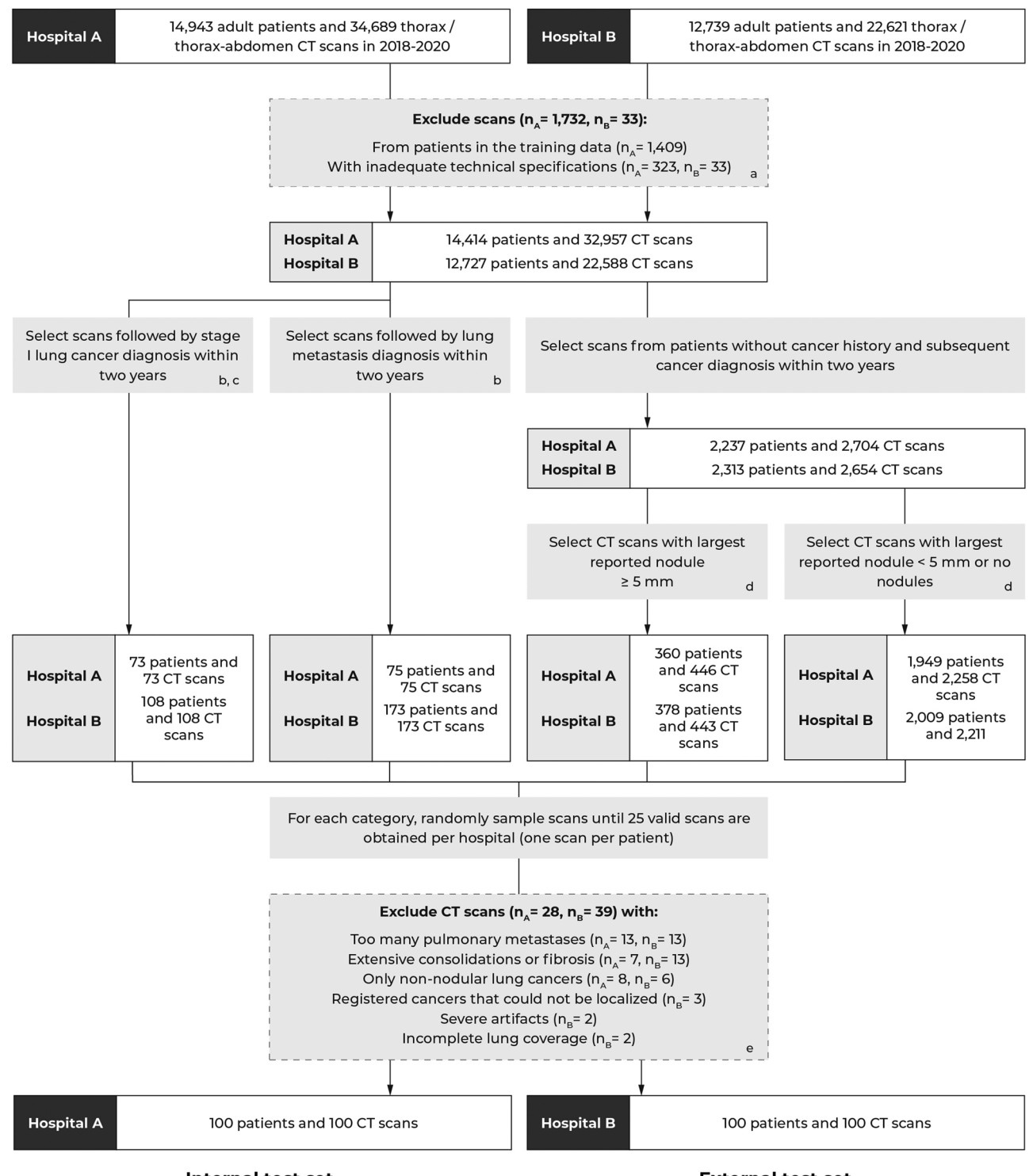

**Fig. 1 Flowchart for creating the test dataset for the evaluation of the pulmonary nodule detection system.** [a]CT scans with thick slices (>3 mm), missing slices, or very low volume (<50 slices) were excluded. [b]The most recent CT scan prior to the cancer diagnosis was selected to ensure retrospective localization. [c]Only selected primary lung cancers that were histologically examined. [d]Based on a natural language processing analysis of the radiology reports[51]. [e]One CT scan per patient was selected. Non-nodular lung cancers were cancers that did not appear as nodules (i.e., masses, thick-walled cysts).

Characteristics of all pulmonary nodules in the training and test datasets are described in Table 2 and additional information about the training labels is included in Supplementary Table 3. The characteristics of the subset of malignant nodules in the test datasets are provided in Supplementary Table 4.

**Eligibility criteria**. In accordance with the British Thoracic Society (BTS) nodule management guidelines, only adult patients (≥18 years old) were included[26]. For the selection primary lung cancer cases, we included patients with stage I cancer as they include nodules instead of masses (>30 mm)[27]. For the selection

of pulmonary metastases cases, both patients with metastasized lung cancer and extra-thoracic cancer were included.

Considering the routine clinical setting of our study, it is important to note that not all patients can be reliably screened for malignant pulmonary nodules. Patients with extensive fibrosis or consolidations (e.g., due to severe interstitial diseases, hemorrhage, or pneumonia) were excluded, as their lungs contain high attenuation areas that prevent correct location and delineation of relevant nodules. Furthermore, patients were excluded if CT scans were made with a slice thickness >3 mm, or were limited by severe breathing artifacts or incomplete coverage of the lungs. Finally, patients with more than 15 pulmonary metastases (according to the initial visual assessment) were excluded to reduce annotation efforts and prevent data imbalance.

**Reference standard**. A panel of five thoracic radiologists (J.T.-d.J., S.S., M.M., M.v.L., M.K. with 2, 4, 6, 16, 21 years of experience, respectively) independently annotated and measured all intra-pulmonary nodules in the test datasets with in-house software (version 19.9.2 of CIRRUS Lung Screening, DIAG, Radboudumc, Nijmegen, The Netherlands). Nodules were manually identified and then volumetrically measured using a semi-automatic nodule segmentation algorithm[28]. Radiologists were able to manually correct nodule segmentations during this process. Furthermore, they indicated the lobe location and type of the nodules (solid, part-solid, non-solid, perifissural, and calcified). The radiologists were instructed to annotate all intrapulmonary nodules, defined as any round or irregular density inside the lung parenchyma with an equivalent diameter ≥3 and ≤30 mm[29]. We matched the nodule annotations of the different radiologists and used a majority vote reference standard that only included the nodules that were detected by at least three radiologists. The remaining annotations (i.e., lesions found to be <3 mm or >30 mm or nodules annotated by the minority of radiologists) were considered as indeterminate findings and were moved to an exclusion list, which is consistent with the reference standard from the LUNA16 challenge (more details in section Analysis).

Two radiologists (E.T.S. and M.R., not part of the panel) linked the annotated nodules to the cancer diagnoses and checked for any missed cancers. They were provided with all available CT scans (period 2000–2020) of a patient; the corresponding radiology reports; and the lobe location, nodule diameter, and histological type of the primary and metastasized cancers as recorded in the NCR. All cancer diagnoses were either confirmed by histological examination, cytology testing, or clinical diagnostic testing (e.g., medical imaging, exploratory surgery). The basis of all cancer diagnoses and cancer morphology are provided in Supplementary Tables 5 and 6. A lesion was considered benign if it was stable and not followed by a cancer diagnosis within two years, although this does not completely rule out the possibility of an indolent malignancy in a stable subsolid nodule[30].

**Nodule detection pipeline**. An overview of the nodule detection pipeline is displayed in Fig. 2. The pulmonary nodule detection system consists of three components that each use deep learning architectures for the following tasks: (1) lung detection, (2) nodule candidate detection, and (3) false positive reduction. A detailed description of the design of these components and the training procedure is provided in Supplementary Note 2.

The lung and nodule candidate detection models are one-stage 2D object detectors with the YOLOv5 architecture (version 5.0, 2021)[31]. The slice-by-slice 2D lung bounding boxes are combined into a 3D volume of interest. This preprocessing step enables a fast and accurate localization of the lungs, and thereby reduces the computational load for the subsequent components, especially for the analysis of larger CT scans that contain both the thorax and abdomen. The nodule candidate detection component is designed to detect potential nodule locations with the highest possible sensitivity. As in previous work[32–34], this component uses consecutive axial CT slices as input channels, thereby adding additional spatial information to the single 2D input images. This procedure helps to discriminate nodules from pulmonary vessels and other linear structures.

The false positive reduction component reduces the number of false positives while retaining a high sensitivity. The false positive reduction model is adapted from the work of Venkadesh et al.[35], which is a multi-view ResNet50 classification model that takes nine different slices from a 3D patch around a nodule candidate. The nodule detection pipeline returns the center coordinates of the detected nodules and their nodule likelihood scores.

**Analysis**. The nodule detection system was evaluated on the internal and external test set by measuring the sensitivity and corresponding false positive rate per scan on different operating points on the Free Receiver Operating Characteristic (FROC) curve. We assessed the sensitivity at 7 predefined false positive rates, namely 0.125, 0.25, 0.5, 1, 2, 4, and 8 false positives per scan. We also assessed the average sensitivity at all false positive rates, referred to as the Competition Performance Metric (CPM) in the LUNA16 challenge[25]. For each threshold, the 95% confidence interval was calculated by using bootstrapping (1,000 bootstraps using scan-level sampling with replacement). These evaluation metrics were calculated with the Python scripts that were provided for the LUNA16 challenge[25]. We evaluated the nodule detection system for multiple subsets of nodules with a minimum diameter threshold of 3, 4, and 5 mm. Additional

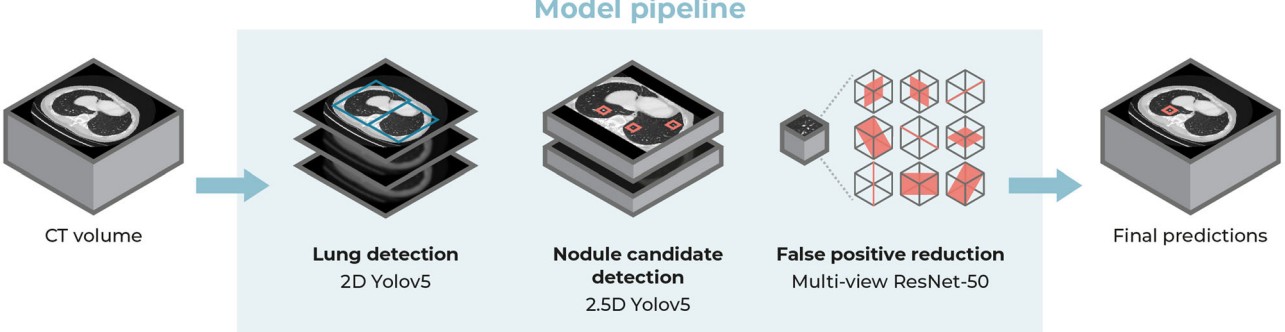

**Fig. 2 Components of the pulmonary nodule detection system.** First, the system takes a CT scan and detects the lungs slice-by-slice to obtain a region of interest. Second, nodule candidates are generated by analyzing overlapping CT volumes of five slices each. Finally, nine different 2D views are sampled from each nodule candidate and analyzed by a multi-view ResNet-50 network.

analyses were conducted for primary lung cancers, pulmonary metastases, and actionable benign nodules (≥5 mm, neither calcified nor perifissural), which would require follow-up according to the BTS guidelines[26].

Our hit criterion was that the center of a predicted nodule should be within the radius of the ground truth nodule, otherwise a detection was regarded as a false positive. If a nodule prediction matched with a nodule regarded as indetermined (see section "Reference standard"), then it was ignored and not counted as a true or false positive. To characterize failure modes of the AI system, we asked an experienced thoracic radiologist (E.T.S.) to perform a visual inspection of all false negative detections and 25 randomly sampled false positive detections of the nodule detection system for each test set. For this analysis, we selected a clinically acceptable operating point for the model that corresponded to an average of 1 FP/s.

To assess the potential clinical value of the proposed AI system, its sensitivity for detecting primary cancers, pulmonary metastases, and actionable benign nodules was compared to those from the individual radiologists from the panel. The sensitivity and false positive rate of each radiologist were estimated by comparing his or her annotations with a new set defined by the other four radiologists (for the definition of benign and indetermined nodules). Since the reference standard slightly changed for each radiologist, the sensitivity of the AI system at 1 FP/s was recalculated for each comparison. The 95% confidence intervals for the AI system and radiologists were calculated by using bootstrapping (1000 bootstraps using scan-level sampling with replacement). Significance testing was performed with a two-sided paired permutation test (1000 iterations) on nodule-level with the MLxtend library (version 0.22.0, 2023)[36] for Python. A $p$ value smaller than 0.05 was considered significant.

**Reporting summary**. Further information on research design is available in the Nature Portfolio Reporting Summary linked to this article.

## Results

**Test datasets characteristics**. In the period of 2018-2020, 14,943 adult patients underwent 34,689 thorax and thorax–abdomen CT scans at hospital A (see Fig. 1). In hospital B, 12,739 patients underwent 22,621 thorax and thorax–abdomen CT scans in the same period. From these scans, 356 scans ($n_A = 323$, $n_B = 33$) were excluded due to inadequate technical specifications (e.g., thick slices, missing slices, or very low volume). Initial samples of 2852 (hospital A) and 2935 (hospital B) CT scans were drawn from four nodule categories for the respective internal and external test set. For each category, scans were randomly sampled until 25 valid scans were obtained per hospital. During this process, 267 scans ($n_A = 128$, $n_B = 139$) were assessed. In total, 67 scans ($n_A = 28$, $n_B = 39$) were excluded as they had too many pulmonary metastases ($n_A = 13$, $n_B = 13$), extensive consolidations or fibrosis ($n_A = 7$, $n_B = 13$), only non-nodular lung cancers (e.g., masses) ($n_A = 8$, $n_B = 6$), registered cancers that could not be retrospectively localized ($n_B = 3$), insufficient lung coverage ($n_B = 2$), or severe artifacts ($n_B = 2$). This resulted into a final selection of 100 studies from 100 patients (63 ± 15 years, 52 women) for the internal test set (hospital A) and 100 studies from 100 patients (67 ± 12 years, 53 women) for the external test set (hospital B) (see Table 1).

For these test sets, 622 ($n_A = 319$, $n_B = 303$) from the 1,617 annotations ($n_A = 852$, $n_B = 765$) in total were included in the analysis (see Table 2). The remaining findings were considered as indetermined, as they did not meet the size criteria ($n_A = 218$, $n_B = 157$) or were non-cancerous and not labeled by the majority of radiologists ($n_A = 315$, $n_B = 305$).

**Pulmonary nodule detection analysis**. Table 3 presents the sensitivity of the AI system at 0.125, 0.25, 0.5, 1, 2, 4, and 8 false positives per scan (FP/s) on the internal and external test set. The FROC curves for the detection of actionable benign nodules, primary lung cancers, and pulmonary metastases per test set are shown in Fig. 3. Supplementary Note 4 describes additional analyses of the detection performance of the individual components of the AI system. More specifically, data characteristics and results of the lung detection component are presented in Supplementary Tables 7 and 8. Performance of the nodule candidate detection and false positive reduction components are provided in Supplementary Table 9. Nodule detection results on subgroups of contrast-enhanced and non-contrast CT scans are provided in

### Table 1 Characteristics of the training and test datasets.

| Dataset | Lung detection training dataset | Nodule detection training dataset | Internal test dataset | External test dataset |
|---|---|---|---|---|
| Source(s)[a] | Hospital A + LIDC/IDRI | Hospital A + LIDC/IDRI | Hospital A | Hospital B |
| Period[b] | 2011, 2017 | 2011, 2017 | 2018–2020 | 2018–2020 |
| Patients ($n$, % of all patients) | | | | |
| All | 1387 | 1489 | 100 | 100 |
| Men | 273 (19.7) | 327 (22.0) | 48 (48.0) | 47 (47.0) |
| Women | 227 (16.4) | 275 (18.5) | 52 (52.0) | 53 (53.0) |
| N/A[b] | 887 (64.0) | 887 (59.6) | | |
| Age (mean, SD)[b] | | | | |
| All | 59.6 (15.3) | 61.7 (14.0) | 62.5 (15.0) | 66.9 (11.7) |
| Men | 58.8 (15.8) | 62.2 (14.3) | 61.8 (16.3) | 66.0 (13.3) |
| Women | 60.6 (14.6) | 61.1 (13.5) | 63.1 (13.8) | 67.8 (10.2) |
| CT scans ($n$, % of all scans) | | | | |
| All | 1388 | 1490 | 100 | 100 |
| Contrast-enhanced | 605 (43.6) | 508 (34.1) | 68 (68.0) | 67 (67.0) |
| Slice thickness in mm (mean, range) | 1.44 (0.50–3.00) | 1.38 (0.25–2.50) | 0.67 (0.25–3.00) | 2.47 (0.6–3.0) |
| Axial plane resolution in mm (mean, range) | 0.70 (0.30–1.02) | 0.67 (0.29–0.98) | 0.64 (0.31–0.93) | 0.73 (0.53–0.98) |

[a]We selected the same subset of CT scans as used in the LUNA16 challenge[25]. Note that one patient had two CT scans.
[b]Patient information (i.e., age and sex) and study dates are unavailable for the LIDC-IDRI dataset due to anonymization[14]. The LIDC-IDRI dataset was released in 2011.

Supplementary Table 10. The average processing time per scan was $30 \pm 18$ s.

For the internal test set, the sensitivity for detecting all nodules at an average of 1 FP/s (detection threshold = 0.647) was 90.9% (290/319, 95% CI: 88.0–93.6%) and the CPM was 85.4%. For detecting actionable benign nodules, the sensitivity at 1 FP/s was 92.1% (58/63, 95% CI: 84.3–98.4%) and the CPM was 90.9%. For detecting primary lung cancers, the sensitivity at 1 FP/s was 92.6% (25/27, 95% CI: 82.1–100%) and the CPM was 91.5%. For detecting pulmonary metastases, the sensitivity at 1 FP/s was 90.3% (149/165, 95% CI: 85.6–94.5%) and the CPM was 83.9%.

For the external test set, the sensitivity for detecting all nodules at an average of 1 FP/s (detection threshold = 0.573) was 92.4% (280/303, 95% CI: 89.8–95.1%) and the CPM was 87.6%. For detecting actionable benign nodules, the sensitivity at 1 FP/s was 94.3% (82/87, 95% CI: 88.1–98.8%) and the CPM was 90.5%. For detecting primary lung cancers, the sensitivity at 1 FP/s was 96.9% (31/32, 95% CI: 91.7–100%) and the CPM was 94.2%. For detecting pulmonary metastases, the sensitivity at 1 FP/s was 92.0% (104/113, 95% CI: 88.5–95.5%) and the CPM was 89.0%. The sensitivity of the nodule detection system for the nodules with a minimum diameter threshold of 4 and 5 mm are included in Table 3.

**Visual assessment of false positives and false negatives**. All false negatives ($n_A = 29$, $n_B = 23$) and a random selection of false positives ($n_A = 25$, $n_B = 25$) from the AI system at an operating point of 1 FP/s were visually assessed. The false negatives could be divided into seven categories and the false positives into fourteen categories, as outlined in Fig. 4. For the false negative categories, it is also shown how many nodules missed by the AI model were still detected by the radiologists from the panel for reference purposes. The three most frequent false negatives were juxta-pleural nodules ($n_A = 17$, $n_B = 11$; defined as solid nodules located on or within 10 mm of the visceral pleura[37]), juxtavas-cular nodules ($n_A = 5$, $n_B = 2$, defined as solid nodules that are attached to a vessel), and non-solid nodules ($n_A = 3$, $n_B = 3$). Regarding the false positives, the three most frequent false categories were fibrosis ($n_A = 2$, $n_B = 4$), duplicate nodule detections ($n_A = 2$, $n_B = 3$), and consolidations ($n_A = 2$, $n_B = 1$). Potentially missed solid nodules ($n_A = 7$, $n_B = 7$), perifissural nodules ($n_A = 2$, $n_B = 3$), and micronodules ($n_A = 1$, $n_B = 4$; defined as nodules smaller than 3 mm[29]) were counted as false positives in accordance with our reference standard.

**Comparison of AI performance with the panel of radiologists**. Figure 5 shows examples of malignant nodules that were missed

**Table 2 Characteristics of the pulmonary nodules in the training and test datasets.**

| Dataset | Nodule detection training dataset | Internal test set | External test set |
|---|---|---|---|
| Source(s) | Hospital A + LIDC/IDRI | Hospital A | Hospital B |
| Total nodules | 4770 | 319 | 303 |
| Nodules per diameter threshold (n, % of total)[a] | | | |
| ≥4 mm | 3576 (75.0) | 250 (78.4) | 262 (86.4) |
| ≥5 mm | 2453 (51.4) | 188 (58.9) | 215 (71.0) |
| Diameter (in mm)[a] | | | |
| Median | 5.0 | 5.5 | 6.6 |
| IQR | 4.0–7.0 | 4.1–8.6 | 4.7–11.9 |
| Volume (in mm³)[a] | | | |
| Median | 68 | 88 | 161 |
| IQR | 33–186 | 36–332 | 55–889 |
| Nodules per scan | | | |
| Median | 2 | 1 | 2 |
| IQR | 1–4 | 1–4 | 1–4 |
| Nodules per type (n, % of total)[b] | | | |
| Solid | 2739 (57.5) | 269 (84.3) | 247 (81.5) |
| Part-solid | 449 (9.4) | 12 (3.8) | 19 (6.3) |
| Non-solid | 469 (9.8) | 8 (2.5) | 10 (3.3) |
| Perifissural | 684 (14.3) | 24 (7.5) | 17 (5.6) |
| Calcified | 429 (9.0) | 6 (1.9) | 10 (3.3) |
| Benign versus malignant nodules (n, % of total)[c] | | | |
| Benign | N/A | 127 (39.8) | 158 (52.1) |
| Primary cancer | N/A | 27 (8.5) | 32 (10.6) |
| Metastasis | N/A | 165 (51.7) | 113 (37.3) |
| Actionable benign nodules (n, % of total) | N/A | 63 (19.7) | 87 (28.7) |

[a]In case of multiple annotations, the volume and equivalent diameter labels from the different readers were averaged per nodule. Interquartile range (IQR) is from the 25th to the 75th percentile.
[b]For the test datasets, the nodule type was based on the majority vote. For the training dataset, the nodule type was determined by either a majority vote (source: LIDC-IDR; value of 2 out of 5 was considered as non-solid, value of 4 out of 5 as part-solid) or single reader (Hospital A). Perifissural nodules were not labeled in the LIDC-IDRI dataset.
[c]Malignancy labels were not available for the training dataset.

**Table 3 Pulmonary nodule detection results on the internal and external test set.**

| | | Average number of false positives per scan | | | | | | | CPM |
|---|---|---|---|---|---|---|---|---|---|
| | Count | 0.125 | 0.25 | 0.5 | 1 | 2 | 4 | 8 | |
| *Internal (hospital A)* | | | | | | | | | |
| All | | | | | | | | | |
| Nodules ≥3 mm | 319 | 60.8 | 72.1 | 82.1 | 90.9 | 95.6 | 97.5 | 99.1 | 85.4 |
| Nodules ≥4 mm | 250 | 72.4 | 80.8 | 89.2 | 92.8 | 96.8 | 98.0 | 99.2 | 89.9 |
| Nodules ≥5 mm | 188 | 77.1 | 83.0 | 89.9 | 91.5 | 96.3 | 97.9 | 99.5 | 90.7 |
| Benign | | | | | | | | | |
| Actionable nodules | 63 | 77.8 | 82.5 | 88.9 | 92.1 | 96.8 | 98.4 | 100.0 | 90.9 |
| Malignant | | | | | | | | | |
| Primary cancers | 27 | 77.8 | 81.5 | 92.6 | 92.6 | 96.3 | 100.0 | 100.0 | 91.5 |
| Metastases | 165 | 60.0 | 68.5 | 78.2 | 90.3 | 95.2 | 96.4 | 98.8 | 83.9 |
| *External (hospital B)* | | | | | | | | | |
| All | | | | | | | | | |
| Nodules ≥3 mm | 303 | 68.6 | 77.3 | 86.5 | 92.4 | 94.4 | 96.7 | 97.4 | 87.6 |
| Nodules ≥4 mm | 262 | 76.3 | 83.6 | 90.1 | 94.3 | 96.2 | 97.3 | 97.7 | 90.8 |
| Nodules ≥5 mm | 215 | 79.1 | 85.1 | 90.7 | 94.4 | 96.7 | 97.7 | 98.1 | 91.7 |
| Benign | | | | | | | | | |
| Actionable nodules | 87 | 72.4 | 82.8 | 90.8 | 94.3 | 96.6 | 97.7 | 98.9 | 90.5 |
| Malignant | | | | | | | | | |
| Primary cancers | 32 | 87.5 | 90.6 | 93.8 | 96.9 | 96.9 | 96.9 | 96.9 | 94.2 |
| Metastases | 113 | 76.1 | 79.8 | 88.5 | 92.0 | 93.8 | 96.5 | 96.5 | 89.0 |

Note. The model sensitivity (%) is reported for each false positive rate.
*CPM* competition performance metric, the average sensitivity at all false positive rates.

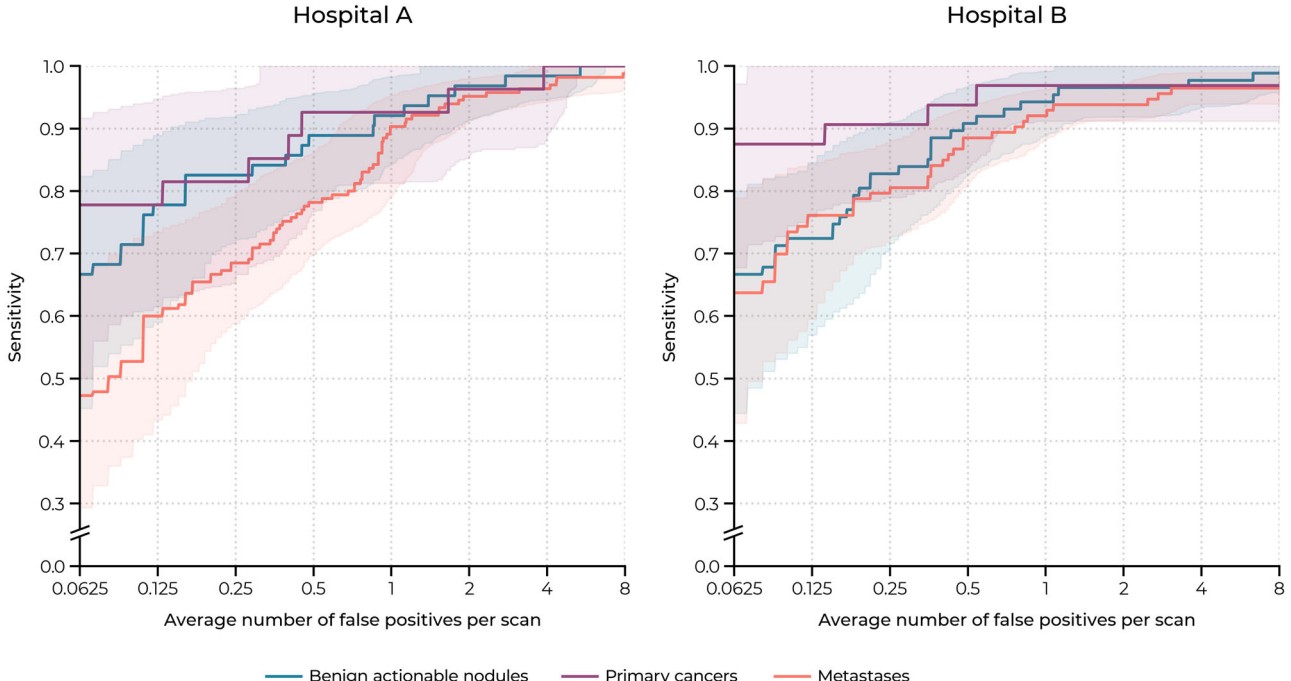

**Fig. 3 Free response receiver operating characteristic (FROC) curves of the AI system per test set.** The internal test set (hospital A) contained 27 primary lung cancers, 165 pulmonary metastases, and 63 actionable benign nodules. The external test set (hospital B) contained 32 primary lung cancers, 113 pulmonary metastases, and 87 actionable benign nodules. The shaded bands represent the 95% confidence intervals per nodule category.

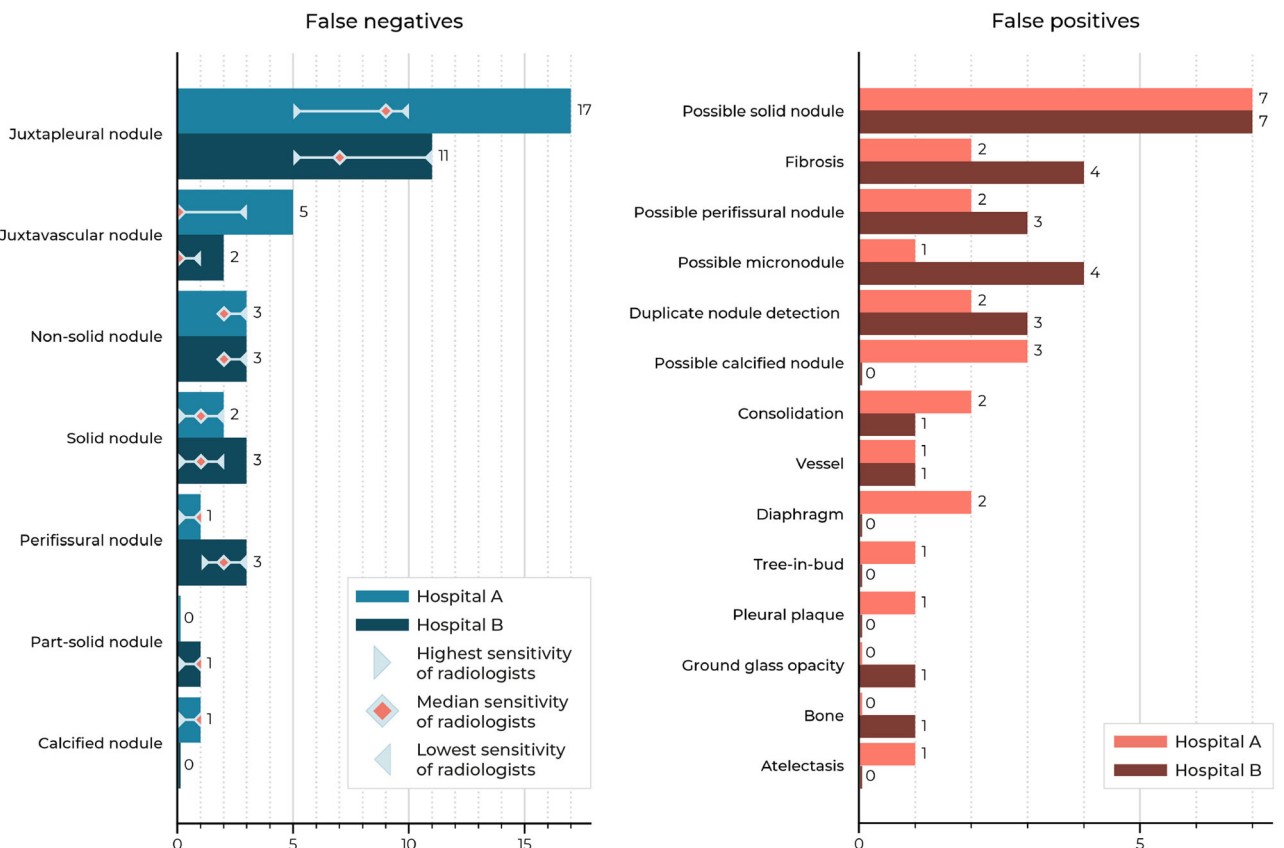

**Fig. 4 Frequency distribution of the categories of the false negative and false positive detections of the AI system.** In the internal test set (hospital A), 29 false negative and 25 false positive detections were inspected. In the external test set (hospital B), 23 false negative and 25 false positive detections were inspected. The lowest, median, and highest sensitivity of the radiologists have been indicated for the false negative detections of the AI system.

**Primary cancers or metastases missed by AI, but not by majority of radiologists**

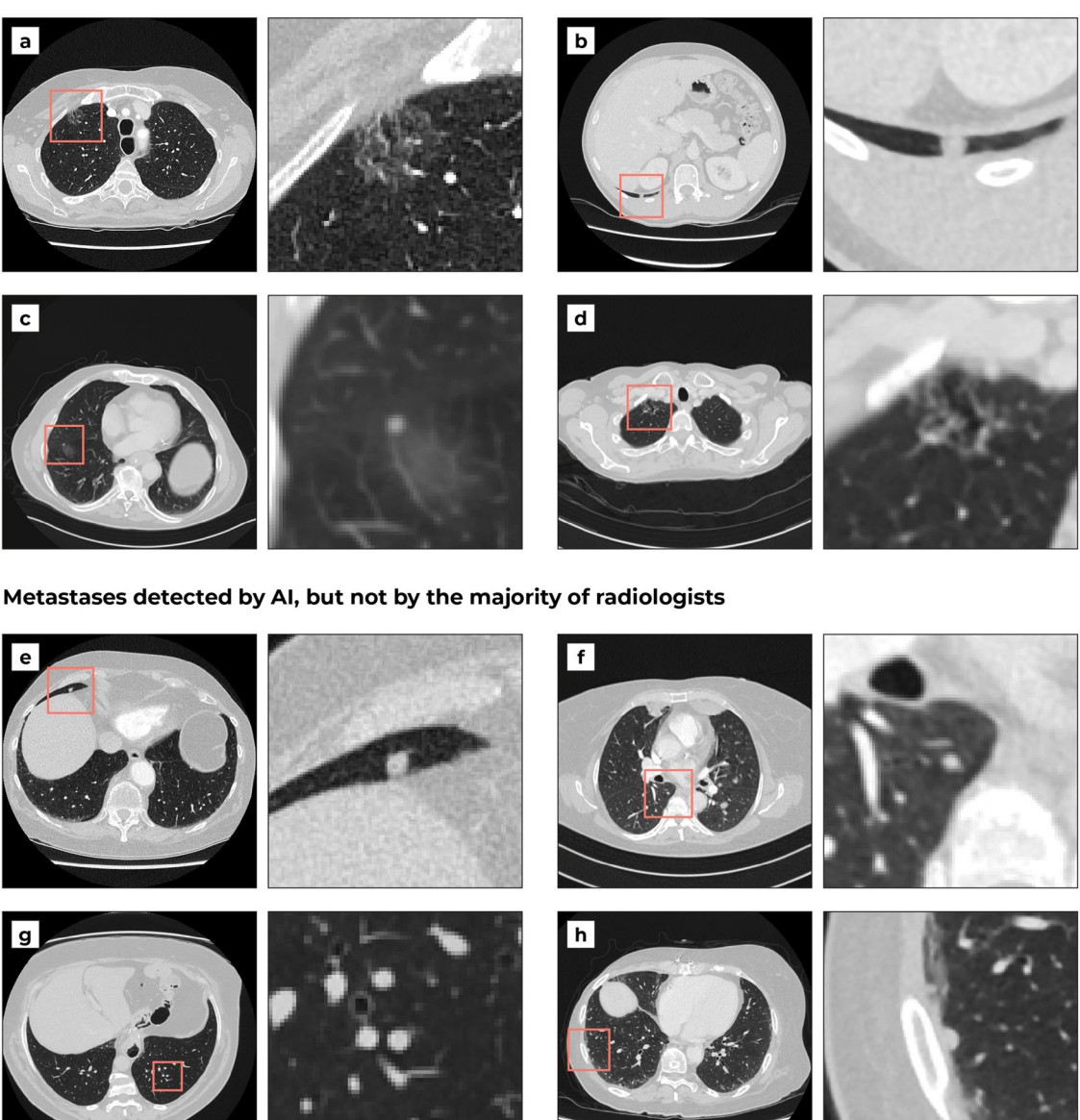

**Metastases detected by AI, but not by the majority of radiologists**

**Fig. 5 Examples of primary cancers and pulmonary metastases missed by the AI system or majority of radiologists from the panel.** The bounding boxes are 60 mm. The most frequent false negatives of the AI model were non-solid nodules (**a**), juxtapleural nodules (**b**, **c**), and part-solid nodules (**d**). In contrast to the AI model, the most frequent false negatives of the radiologists were nodules in base of the lungs (**e**), nodules in the right azygoesophageal recess (**f**), nodules with the same diameter as neighbouring vessels (**g**), and juxtapleural nodules (**h**).

by either the AI system or majority of radiologists in the panel. The majority of radiologists detected all primary lung cancers in the internal and external test set, but the AI system missed 2 out of 27 cancers in the internal test set and 1 out of 32 cancers in the external test set. In the internal test set, 55 out of 165 metastases (33%) were missed by the majority of radiologists (15/25 patients). The AI system detected 41 (75%) of these missed metastases (14/15 patients). For the external test set, 27 out of 113 metastases (24%) were missed by the majority of radiologists (11/25 patients). The AI system detected 21 (78%) of these missed metastases (10/11 patients).

In Table 4, the sensitivity of the AI system for detecting actionable benign nodules, primary lung cancers, and pulmonary metastases at 1 FP/s was compared with the sensitivity of each individual radiologist from the panel. The FROC curve of the AI

system (average of all comparisons) and operating points of the radiologists are visualized in Fig. 6. The AI system had a significantly higher sensitivity than 2 out of 5 radiologists for detecting actionable benign nodules in the internal test set (radiologist 2, 95% vs. 79%, $p = 0.02$; radiologist 4, 95% vs. 79%, $p = 0.03$) and external test set (radiologist 2, 95% vs. 70%, $p < 0.001$; radiologist 4, 95% vs. 66%, $p < 0.001$), although with a higher false positive rate than the radiologists (average difference of 0.6 FP/s). For detecting pulmonary metastases, the AI system had a significantly higher sensitivity than 4 out of 5 radiologists in the internal test set (radiologist 1, 86% vs. 77%, $p = 0.01$; radiologist 2, 90% vs. 32%, $p < 0.001$; radiologist 3, 88% vs. 70%, $p < 0.001$; radiologist 4, 88% vs. 45%, $p < 0.001$) and external test set (radiologist 1, 90% vs. 81%, $p = 0.048$; radiologist 2, 92% vs. 66%, $p < 0.001$; radiologist 3, 92% vs. 75%, $p < 0.001$; radiologist 4, 92%

**Table 4 Comparison between the nodule detection performance of the AI model and individual readers on the internal and external test set.**

| | FP/scan | Actionable benign nodules | | Primary cancers | | Metastases | |
|---|---|---|---|---|---|---|---|
| | | Sensitivity (%) | p | Sensitivity (%) | p | Sensitivity (%) | p |
| **Internal** | | | | | | | |
| Radiologist 1 | 0.7 (0.5, 0.9) | 98 (94, 100) | | 100 (100, 100) | | 77 (66, 88) | |
| Recalibrated AI | 1.0 (0.7, 1.5) | 92 (84, 99) | 0.24 | 93 (82, 100) | 0.50 | 86 (81, 92) | 0.01 |
| Radiologist 2 | 0.1 (0.0, 0.2) | 79 (69, 89) | | 100 (100, 100) | | 32 (18, 47) | |
| Recalibrated AI | 1.0 (0.7, 1.5) | 95 (88, 100) | 0.02 | 93 (82, 100) | 0.52 | 90 (85, 94) | <0.001 |
| Radiologist 3 | 0.2 (0.1, 0.3) | 89 (81, 97) | | 96 (87, 100) | | 70 (57, 83) | |
| Recalibrated AI | 1.0 (0.7, 1.5) | 91 (83, 98) | >0.99 | 93 (82, 100) | >0.99 | 88 (84, 93) | <0.001 |
| Radiologist 4 | 0.5 (0.3, 0.7) | 79 (70, 88) | | 100 (100, 100) | | 45 (37, 54) | |
| Recalibrated AI | 1.0 (0.7, 1.5) | 95 (88, 100) | 0.03 | 93 (82, 100) | 0.52 | 88 (84, 93) | <0.001 |
| Radiologist 5 | 0.8 (0.5, 1.0) | 77 (59, 95) | | 100 (100, 100) | | 87 (80, 93) | |
| Recalibrated AI | 1.0 (0.7, 1.5) | 89 (79, 97) | 0.08 | 93 (82, 100) | 0.50 | 83 (77, 89) | 0.23 |
| **External** | | | | | | | |
| Radiologist 1 | 0.7 (0.4, 1.0) | 90 (81, 96) | | 97 (90, 100) | | 81 (74, 91) | |
| Recalibrated AI | 1.0 (0.7, 1.5) | 97 (91, 100) | 0.16 | 97 (92, 100) | >0.99 | 90 (86, 95) | 0.048 |
| Radiologist 2 | 0.1 (0.1, 0.2) | 70 (60, 78) | | 91 (78, 100) | | 66 (59, 73) | |
| Recalibrated AI | 1.0 (0.7, 1.4) | 95 (89, 99) | <0.001 | 97 (92, 100) | 0.63 | 92 (89, 96) | <0.001 |
| Radiologist 3 | 0.3 (0.1, 0.4) | 94 (86, 99) | | 94 (86, 100) | | 75 (65, 87) | |
| Recalibrated AI | 1.0 (0.6, 1.4) | 95 (89, 100) | >0.99 | 97 (92, 100) | >0.99 | 92 (89, 96) | <0.001 |
| Radiologist 4 | 0.5 (0.4, 0.7) | 66 (53, 78) | | 97 (90, 100) | | 61 (50, 74) | |
| Recalibrated AI | 1.0 (0.7, 1.4) | 95 (89, 99) | <0.001 | 97 (92, 100) | >0.99 | 92 (89, 96) | <0.001 |
| Radiologist 5 | 0.6 (0.4, 0.7) | 86 (75, 95) | | 97 (90, 100) | | 88 (83, 93) | |
| Recalibrated AI | 1.0 (0.6, 1.3) | 96 (89, 100) | 0.07 | 97 (92, 100) | >0.99 | 91 (87, 95) | 0.64 |

Note. The AI system was recalibrated for each comparison to match an average FP/s of 1, as the exclusion list and reference standard slightly varied per reader (latter only for benign nodules). The 95% confidence intervals are enclosed by parentheses.
*FP/scan* average number of false positives per scan.

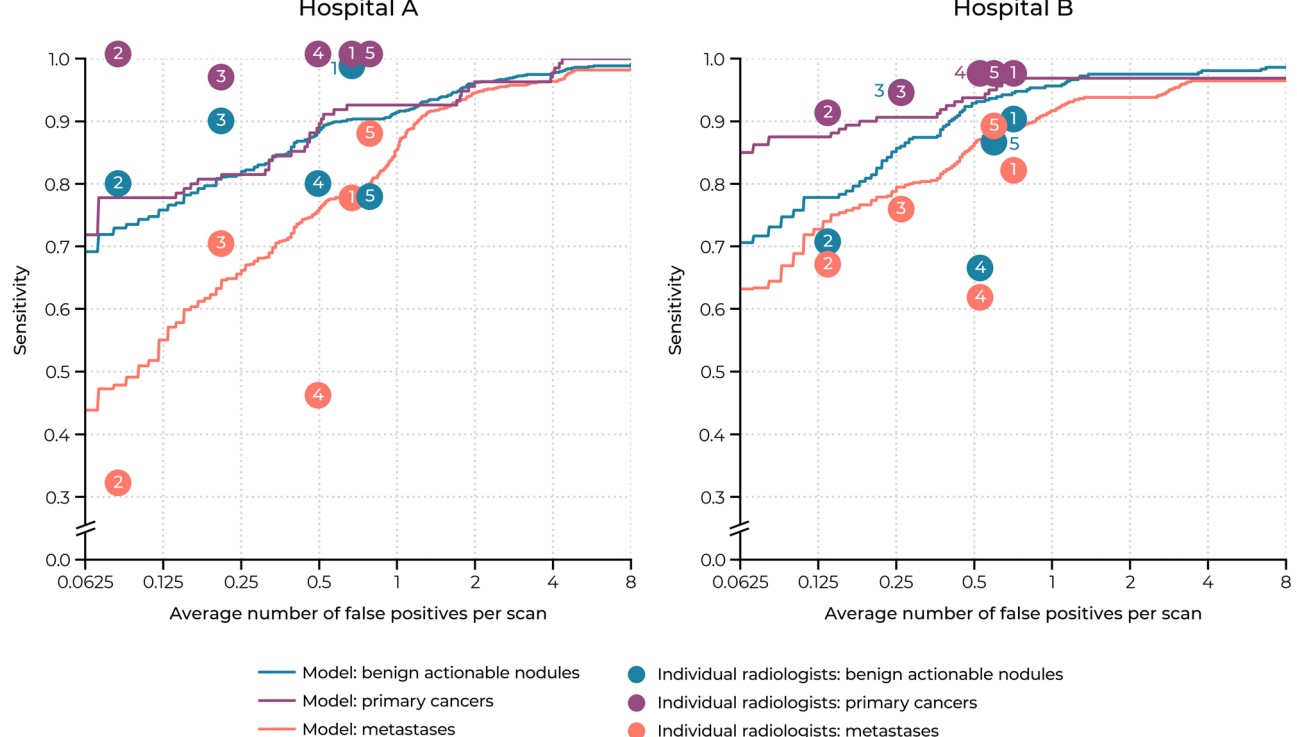

**Fig. 6 Operating points of the individual radiologists of the panel compared to the free response receiver operating characteristic (FROC) curves of the AI system per test set.** The FROC curves are averaged over all comparisons per nodule category.

vs. 61%, *p* < 0.001). There was no significant difference between the sensitivity of the AI system and the radiologists for detecting primary lung cancer. In all other cases, there was no significant difference in sensitivity between the AI system and radiologists.

## Discussion
In recent years, DL-CAD systems have shown a high performance for detecting pulmonary nodules in publicly available CT datasets. However, many nodule detection systems have been

neither externally validated in a clinical setting nor specifically validated for the detection of cancer. For these reasons, we developed and evaluated a DL-CAD system for the detection of pulmonary nodules in routine clinical CT scans with a known malignancy status. At a clinically acceptable threshold of 1 FP/s, the system obtained a sensitivity of 92% for detecting nodules with a minimum diameter of 3 mm on the external test set. For actionable benign nodules, primary lung cancers and pulmonary metastases, the sensitivity was 94%, 97%, and 92%, respectively. The detection performance for hospitals A and B was comparable, indicating a good generalization performance. A comparison between the nodule detection performance of the DL-CAD system and radiologists showed that the system could provide a higher sensitivity (average difference of 14, 2, and 17 percentage points for actionable benign nodules, primary lung cancers and pulmonary metastases, respectively) at the cost of a slightly higher false positive rate (average difference of 0.6 FP/s), and that it could locate most pulmonary metastases (78%) that were missed by the majority of the five radiologists.

The sensitivity of our system for detecting pulmonary nodules is comparable to the sensitivities reported in previous evaluation studies of DL-CAD systems in a non-screening setting. Studies have reported sensitivities in the range of 65–88% at 1 FP/s for pulmonary nodules of any diameter[38–41]. For nodules with a minimum diameter of 3, 4, and 5 mm, detection sensitivities are reported of 74%[33], 88%[42], and 82–91%[15,43] at 1 FP/s, respectively. In future research, DL-CAD systems should be benchmarked on a modern dataset with routine clinical CT scans to determine the most optimal detection method for analyzing these highly heterogenous scans.

To the best of our knowledge, there have been no studies in recent years that evaluated a DL-CAD system for the detection of both small lung cancers and pulmonary metastases in routine clinical CT scans. For case-level lung cancer detection however, Zhang et al.[24] have shown that their DL-CAD system can obtain a sensitivity of 96% and specificity of 88% on a dataset of 50 preoperative CT scans, from which half contained pathologically confirmed malignant nodules. This sensitivity is in agreement with our findings, although the specificity cannot be directly compared as our system does not distinguish between benign and malignant nodules.

The assessment of the false negatives and false positives of the DL-CAD system shows that juxtapleural nodules are most challenging to detect, but also that the system can potentially identify nodules that are even missed by a panel of experienced radiologists. Juxtapleural nodules might be hard to detect by the DL-CAD system due to their highly variable shape and similar density to the pleural wall. Besides juxtapleural nodules, the DL-CAD system may miss non-solid nodules, most likely due to their low contrast resolution and the small proportion in the training data as compared to solid nodules (Table 2). The missed primary lung cancers were either non-solid ($n = 2$) or juxtapleural ($n = 1$). Regarding the false positives, most could be considered as possibly overlooked or misinterpreted as non-nodular (13/25 for the internal test set and 14/25 for the external test set). They were often small (<4 mm), had typically benign features (i.e., triangular shape), or were attached to the pleura or vasculature. Previous studies have shown that CAD systems can indeed detect nodules that are missed by multiple readers[39,44]. Other false positive detections were duplicate detections (e.g., nodular components of a larger lesion), non-nodular lesions (e.g., bandlike), fibrosis, and consolidations. Similar causes for false positive findings have been reported by Martins Jarnalo et al.[42].

The comparison between the DL-CAD system and radiologists showed that the system had a significantly higher sensitivity for pulmonary metastases than most radiologist (4 out of 5) and a significantly higher sensitivity for actionable benign nodules than

some radiologists (2 out of 5). No significant differences were found for the detection of primary lung cancer, although the operating points of the radiologists were located above the FROC curve of the DL-CAD system (see Fig. 6). None of the primary lung cancers were missed by the majority of radiologists, probably due to their relatively large size (median diameter of 18 mm in the external test set). The detection of pulmonary metastases was a more challenging task for the radiologists, as these lesions were much smaller (median diameter of 7 mm in the external test set) and appeared in greater numbers (median of four lesions per scan versus one). The difficulty of this task has been demonstrated before in a study of Chen et al., who showed that one or more pulmonary metastases were missed in 37% of all cases[45]. In our datasets, missed pulmonary metastases were usually located in the base of the lower lobes, right azygoesophageal recess, or nearby blood vessels with a similar diameter. These locations are known to be blind spots for radiologists[45,46] and our results suggest that these could be overcome with the help of a DL-CAD system.

Although it has not been demonstrated in this study that a DL-CAD system could improve the radiologists' performance for lung cancer detection, it is likely that the system could aid in the detection of small lung cancers given its high sensitivity for pulmonary metastases and actionable benign nodules. Furthermore, it is important to emphasize that the panel of radiologists were instructed to detect any pulmonary nodule, while this task is not necessarily the focus of a CT examination in daily clinical practice. As a result, certain cognitive biases were less likely to occur in our setting, such as satisfaction of search[11], and the performance of the radiologists might be overestimated.

The strengths of our study are the use of data from routine clinical CT scans from multiple hospitals and a reference standard set by a panel of thoracic radiologists with nodule-level malignancy labels. However, this study also has a few limitations. First of all, we did not conduct a second reading round where the radiologists could review each other's marks, such as the annotation process of the LIDC-IDRI database[14]. More hard-to-detect nodules could have been added to the test datasets by implementing such an annotation process. Secondly, we selected an operating point for the DL-CAD system that matched 1 FP/s to compare its sensitivity with those of the radiologists. This operating point may not necessarily be the optimal trade-off between the sensitivity and false positive rate in terms of costs and benefits and the radiologists' preferences. The selection of optimal operating points should be further investigated. Finally, the CT scans from hospital B were reconstructed with a relatively high slice thickness of 3 mm. This study could have benefited from another external validation set with CT scans with thinner slices.

In conclusion, this study demonstrates that a DL-CAD system obtained a high sensitivity with an acceptable false positive rate for the detection of benign actionable nodules, primary lung cancers, and pulmonary metastases in CT scans from a retrospective cohort of a routine clinical population. The system thereby shows potential for aiding radiologists in detecting small lung cancers and pulmonary metastases for obtaining a timely diagnosis or monitoring disease progression. Future research should focus on the evaluation and implementation of this system in a prospective clinical setting.

## Data availability

Data from the LUNA16 challenge is available via the Cancer Imaging Archive (source data: https://doi.org/10.7937/K9/TCIA.2015.LO9QL9SX[47]; standardized annotations: https://doi.org/10.7937/TCIA.2018.h7umfurq[48]) and Zenodo (part 1: https://doi.org/10.5281/zenodo.3723329[49]; part 2: https://doi.org/10.5281/zenodo.4121926[50]). Clinical data collected at the Radboud University Medical Center and Jeroen Bosch Hospital are not released publicly, but can be requested from the investigators. Reasonable requests for de-identified data for research purposes will be considered by the corresponding author and

requires approval from the institutional review boards before access. Numerical results underlying the graphs in Figs. 3 and 6 are available in Supplementary Data 1 and Supplementary Data 2, respectively.

## Code availability

The proposed system is freely available for research purposes at the platform Grand-Challenge (https://grand-challenge.org/algorithms/lung-nodule-detector-for-ct/). The lung and nodule candidate detection models are based on the YOLOv5 architecture (version 5.0, https://doi.org/10.5281/zenodo.4679653[31], available on Github: https://github.com/ultralytics/yolov5). The nodule false positive reduction model was adapted from the previous work from our group (https://doi.org/10.1148/radiol.2021204433[35]). The model implementation details are described in Supplementary Material. Code for the Free Receiver Operating Characteristic (FROC) analysis is available at the LUNA16 challenge website (https://luna16.grand-challenge.org/Evaluation/). Significance testing was performed with the MLxtend library for Python (version 0.22.0, https://doi.org/10.21105/joss.00638[36]).

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

## Acknowledgements
This work was supported by the Junior Researcher grant from the Radboud Institute for Health Sciences, Radboudumc, Nijmegen, the Netherlands, and the Jeroen Bosch Hospital, Den Bosch, the Netherlands. We thank the registration team of the Netherlands Comprehensive Cancer Organization (IKNL) for the collection of data for the Netherlands Cancer Registry as well as IKNL staff for scientific advice. We acknowledge the National Cancer Institute (NCI) for their publicly available LIDC-IDRI archive. We thank Karlijn Rutten, Noa Antonissen, and Jan Dammeier for their help in the annotation work. We thank Tijs Samson for his support in the data acquisition from the hospital information systems. We thank Kiran Vaidhya Venkadesh for his contribution to the development of the pulmonary nodule detection system.

## Author contributions
W.H. designed and conducted the experiments, analyzed the data, and wrote the manuscript. C.J., M.R., B.v.G., and M.P. supervised the project. C.J. helped with the data acquisition process. N.H. contributed to discussions about the experimental design and data analyses. W.H., M.M., J.T.-d.J., S.S., M.K., M.v.L., E.T.S., and M.R. contributed to the annotation process of the datasets. E.T.S. and M.R. helped with the interpretation of the results. All authors reviewed the manuscript.

## Competing interests
The authors declare the following competing interests: B.v.G. is shareholder and co-founder of Thirona. He reports no other relationships that are related to the subject matter of the article. M.P. receives grants from Canon Medical Systems, Siemens Healthineers; royalties from Mevis Medical Solutions; payment for lectures from Canon Medical Systems, Siemens Healthineers. The host institution of M.P. is a minority shareholder in Thirona. He reports no other relationships that are related to the subject matter of the article. The host institution of C.J. receives research grants and royalties from MeVis Medical Solutions, Bremen Germany. C.J. is a collaborator in a public-private research project where Radboudumc collaborates with Philips Medical Systems (Best, the Netherlands). He reports no other relationships that are related to the subject matter of the article. The other authors declare no competing interests.
