## [Peer Review File · Communications Medicine]

Reviewers' comments:

Reviewer #1 (Remarks to the Author):

Authors proposed a deep learning (DL) based framework for automated detection of lung cancers in Chest CT scans. The proposed framework consists of 3 stages: (1) Lung parenchyma detection, (2) Nodule candidate detection, (3) False positive reduction. The authors combined public dataset with hospital A for model training and hyperparameters tuning, while hospital B used mainly for external set. The study is sound to start with, but I do have a few comments for the authors.

My positive points are:

- The dataset size is a decent number to train a DL-based model to derive a meaningful finding
- Authors took care of data harmonization between multi-centers data and even performed subgroup analysis on certain parameters such as slice thickness and show how the proposed model performed on different CT protocol and settings, which is great.

• Authors balanced the cohort between different types, size of the cancers across different centers
My concerns are as appended:

- Were all the scans contrast-enhanced or a mixed between enhanced and non-enhanced? If the model was trained on a mixed cohort, I would suggest showing how the model performed on a sub-cohort (subgroup analysis) on contrast-enhanced vs unenhanced CT.
- The model was built based on 2D CT slices (slice-by-slice) using a pretrained network that I believe is trained using natural images for object detection. Stage 1 (lung detection) and Stage 2 (nodule candidate detection) utilized 2D and 2.5D RGB input. There has been multiple advancement in medical image analysis in the 3D space such as the JOHOF package (<https://doi.org/10.1186/s41747-020-00173-2>) for lung lobe detection and nn-Unet (<https://doi.org/10.1038/s41592-020-01008-z>) for lesion segmentation that specifically designed for medical images and using NIFTI files as input. I would be more confident with the proposed method if the authors could show that their proposed model works on par (if not better) in comparison to the mentioned tools.
- Authors mentioned that semi-automated tool by reference [28] was utilized to delineate the tumor after radiologist identified the location of the lesion under interest. How did the authors validate the accuracy of the segmentation done by the tool afterwards? Since the output of the segmentation used as the ground truth, I assume some kind of quantitative measurements or some visual corrections considered in that process. (if so, author may want to consider to convey such information to the readers)

Reviewer #2 (Remarks to the Author):

This is a very well-written and designed study describing the development and validation of a deep learning lung nodule detection system proposing to assist radiologists in identifying actionable lung nodules, primary lung cancers and lung metastases. 3 sequential steps are required: (1) lung detection, (2) nodule detection and (3) false positive reduction. The model was trained on a large dataset of LUNA16 scans and a subset of scans from an academic hospital A and validated on a holdout subset of hospital A and an independent dataset from hospital B (100 each, sampled from categories to achieve balanced datasets: stage 1 lung cancer, metastases, benign pulmonary nodules >5 mm and <5mm. Ground truth was established by two expert radiologists and comparison with radiologists involved 5 other trained radiologists who independently reviewed all scans and segmented all nodules. At a FPR of 1/scan calibration, sensitivity for all 4 categories was excellent >90%. This is a strong study, but the lack of prospective validation, the relatively small validation dataset which excluded 25% of scans for are substantial limitations. While interesting, it is difficult to infer from the data presented whether this model would offer substantial improvements in clinical outcomes.

Major concerns:

1. The recalibration of the model cutoff (Table 4) to compare against the radiologists is somewhat confusing and possibly not the best comparison. They recalibrated to a FP/s = 1, which is higher than any of the radiologists' FP/s rate, so the sensitivities between AI and radiologists are difficult to compare. I think either recalibrating the AI to the FP/s rate of each radiologist would be a fairer comparison (essentially, comparing sensitivity at the same specificity), OR presenting the AUC (cutoff agnostic).
2. Figure 5 is not the optimal way to present this data. A bidirectional bar chart should be used to show hazards/effect sizes/major differences in populations. A fairer comparison of the two populations would just use a standard bar chart, where the data have zero on the left and extend to the right, with different colors for the two groups. I would also like to see, rather than "nodules missed by the majority of radiologists," a marker showing the "most accurate, median, and least accurate radiologist" to better contextualize the AI results compared to the radiologist.
3. Lastly, the main conclusion of the article is: "In conclusion, this study demonstrates that a DL-CAD system obtained an expert-level performance for the detection of benign actionable nodules, primary lung cancers, and pulmonary metastases in CT scans from a retrospective cohort of a routine clinical population." The study did not demonstrate expert-level performance, particularly for detection of primary lung cancer. The results show that the difference between expert and AI was not statistically significant ($p > 0.05$), but this does not prove expert-level performance, which would need to be evaluated with a non-inferiority or equivalency design. Specifically, the decrease in sensitivity (even if non-significant) for detecting early stage cancers is concerning because the end goal of nodule detection is detection of cancers. Any missed benign nodule is actually an overall net positive to long term outcomes.

Minor concerns:

1. There are a few typos throughout the manuscript
2. The emphasis on incidentally identified nodules is very useful, but the main manuscript should provide a description of the total pool of chest CTs the external validation dataset was sampled from. How many scans (coronary scans, PE studies, contrast VS not...) would have been excluded from analysis because they do not meet technical eligibility for radiomic analysis?
3. The balanced datasets with 4 categories are useful for validation purposes but introduce another problem of generalizability which call for prospective validation.
4. Geographic differences in nodule prevalence and etiology are likely to substantially impact the external validity of the model as well.
5. 200 scans were selected for validation to achieve sufficient statistical power but the power calculation and assumptions used are not explicitly described.
6. Test dataset should not be under the results section but in the material and methods section since it explains the data design process.
7. Figures 3 and 4 can be combined as they showcase the same quantitative result.
8. In the supplementary material, the authors claim that the object detection approach is far superior compared to segmentation approaches. There is no evidence to prove this either through citations or previous methods that have proved to be slower than the employed method in the paper.
9. The need for a lightweight and heavy weight YOLOv5 model in the lung detection pipeline and the nodule detection pipeline is vague and needs further explanation.

Reviewers' comments

Reviewer #1 (Remarks to the Author):

Authors proposed a deep learning (DL) based framework for automated detection of lung cancers in Chest CT scans. The proposed framework consists of 3 stages: (1) Lung parenchyma detection, (2) Nodule candidate detection, (3) False positive reduction. The authors combined public dataset with hospital A for model training and hyperparameters tuning, while hospital B used mainly for external set. The study is sound to start with, but I do have a few comments for the authors.

My positive points are:

- The dataset size is a decent number to train a DL-based model to derive a meaningful finding
- Authors took care of data harmonization between multi-centers data and even performed subgroup analysis on certain parameters such as slice thickness and show how the proposed model performed on different CT protocol and settings, which is great.
- Authors balanced the cohort between different types, size of the cancers across different centers

My concerns are as appended:

- Were all the scans contrast-enhanced or a mixed between enhanced and non-enhanced? If the model was trained on a mixed cohort, I would suggest showing how the model performed on a sub-cohort (subgroup analysis) on contrast-enhanced vs unenhanced CT.
- The model was built based on 2D CT slices (slice-by-slice) using a pretrained network that I believe is trained using natural images for object detection. Stage 1 (lung detection) and Stage 2 (nodule candidate detection) utilized 2D and 2.5D RGB input. There has been multiple advancement in medical image analysis in the 3D space such as the JOHOF package (<https://doi.org/10.1186/s41747-020-00173-2>) for lung lobe detection and nn-Unet (<https://doi.org/10.1038/s41592-020-01008-z>) for lesion segmentation that specifically designed for medical images and using NIFTI files as input. I would be more confident with the proposed method if the authors could show that their proposed model works on par (if not better) in comparison to the mentioned tools.
- Authors mentioned that semi-automated tool by reference [28] was utilized to delineate the tumor after radiologist identified the location of the lesion under interest. How did the authors validate the accuracy of the segmentation done by the tool afterwards? Since the output of the segmentation used as the ground truth, I assume some kind of quantitative measurements or some visual corrections considered in that process. (if so, author may want to consider to convey such information to the readers)

Reviewer #2 (Remarks to the Author):

This is a very well-written and designed study describing the development and validation of a deep learning lung nodule detection system proposing to assist radiologists in identifying actionable lung nodules, primary lung cancers and lung metastases. 3 sequential steps are required: (1) lung detection, (2) nodule detection and (3) false positive reduction. The model was trained on a large dataset of LUNA16 scans and a subset of scans from an academic hospital A and validated on a

holdout subset of hospital A and an independent dataset from hospital B (100 each, sampled from categories to achieve balanced datasets: stage 1 lung cancer, metastases, benign pulmonary nodules >5 mm and <5mm. Ground truth was established by two expert radiologists and comparison with radiologists involved 5 other trained radiologists who independently reviewed all scans and segmented all nodules. At a FPR of 1/scan calibration, sensitivity for all 4 categories was excellent >90%. This is a strong study, but the lack of prospective validation, the relatively small validation dataset which excluded 25% of scans for are substantial limitations. While interesting, it is difficult to infer from the data presented whether this model would offer substantial improvements in clinical outcomes.

Major concerns:

- The recalibration of the model cutoff (Table 4) to compare against the radiologists is somewhat confusing and possibly not the best comparison. they recalibrated to a FP/s = 1, which is higher than any of the radiologists' FP/s rate, so the sensitivities between AI and radiologists are difficult to compare. I think either recalibrating the AI to the FP/s rate of each radiologist would be a fairer comparison (essentially, comparing sensitivity at the same specificity), OR presenting the AUC (cutoff agnostic).
- Figure 5 is not the optimal way to present this data. A bidirectional bar chart should be used to show hazards/effect sizes/major differences in populations. A fairer comparison of the two populations would just use a standard bar chart, where the data have zero on the left and extend to the right, with different colors for the two groups. I would also like to see, rather than “nodules missed by the majority of radiologists,” a marker showing the “most accurate, median, and least accurate radiologist” to better contextualize the AI results compared to the radiologist.
- Lastly, the main conclusion of the article is: “In conclusion, this study demonstrates that a DL-CAD system obtained an expert-level performance for the detection of benign actionable nodules, primary lung cancers, and pulmonary metastases in CT scans from a retrospective cohort of a routine clinical population.” The study did not demonstrate expert-level performance, particularly for detection of primary lung cancer. The results show that the difference between expert and AI was not statistically significant ($p > 0.05$), but this does not prove expert-level performance, which would need to be evaluated with a non-inferiority or equivalency design. Specifically, the decrease in sensitivity (even if non-significant) for detecting early stage cancers is concerning because the end goal of nodule detection is detection of cancers. Any missed benign nodule is actually an overall net positive to long term outcomes.

Minor concerns:

- There are a few typos throughout the manuscript
- The emphasis on incidentally identified nodules is very useful, but the main manuscript should provide a description of the total pool of chest CTs the external validation dataset was sampled from. How many scans (coronary scans, PE studies, contrast VS not...) would have been excluded from analysis because they do not meet technical eligibility for radiomic analysis?

- The balanced datasets with 4 categories are useful for validation purposes but introduce another problem of generalizability which call for prospective validation.
- Geographic differences in nodule prevalence and etiology are likely to substantially impact the external validity of the mode as well.
- 200 scans were selected for validation to achieve sufficient statistical power but the power calculation and assumptions used are not explicitly described.
- Test dataset should not be under the results section but in the material and methods section since it explains the data design process.
- Figures 3 and 4 can be combined as they showcase the same quantitative result.
- In the supplementary material, the authors claim that the object detection approach is far superior compared to segmentation approaches. There is no evidence to prove this either through citations or previous methods that have proved to be slower than the employed method in the paper.
- The need for a lightweight and heavy weight YOLOv5 model in the lung detection pipeline and the nodule detection pipeline is vague and needs further explanation.

Response to Referees

We would like to thank the reviewers for their conscientious review and comments. This helped us to significantly improve the manuscript. All comments have been addressed and the revisions are highlighted in red.

REVIEWER #1

1. *“Were all the scans contrast-enhanced or a mixed between enhanced and non-enhanced? If the model was trained on a mixed cohort, I would suggest showing how the model performed on a sub-cohort (subgroup analysis) on contrast-enhanced vs unenhanced CT.”*

The AI model was indeed trained and evaluated on a mixed cohort with both contrast-enhanced and non-contrast CT scans (see Table 1; about two-third of the scans in the test sets were contrast-enhanced). We appreciate the suggestion of the reviewer, considering that pulmonary nodules adjacent to vascular structures can be better identified and delineated in contrast-enhanced CT scans [1]. This may affect the detection performance. We have conducted the subgroup FROC analysis of contrast-enhanced versus non-contrast CT scans for the internal and external test sets and have added the results to Supplementary Note 5 and Supplementary Table 10. We evaluated the detection performance for different nodule sizes (3, 4, and 5 mm), but conducted no separate analyses for actionable benign nodules, pulmonary metastases, and primary cancers, as these subgroups became too small ($n < 20$).

The analysis shows that the model performs well for both contrast-enhanced and non-contrast CT scans. For the internal test set, the sensitivity for detecting all nodules in contrast-enhanced CT scans at an average of 1 FP/s was 89.6% (240/268, 95% CI: 86.6%-92.5%) and the CPM was 84.5%. For non-enhanced CT scans, the sensitivity for detecting all nodules at an average of 1 FP/s was 96.1% (49/51, 95% CI: 90.3%-100.0%) and the CPM was 88.6%. For the external test set, the sensitivity for detecting all nodules in contrast-enhanced CT scans at an average of 1 FP/s was 91.9% (217/236, 95% CI: 88.9%-94.8%) and the CPM was 87.4%. For non-enhanced CT scans, the sensitivity for detecting all nodules at an average of 1 FP/s was 94.0% (63/67, 95% CI: 87.2%-98.7%) and the CPM was 88.1%. The differences between the subgroups were similar for the different nodule sizes. We refer to this analysis in the Results section (see changes in red). The results show that the model seems to perform slightly better on the non-contrast CT scans. However, it is important to note that the scans with metastases were all contrast-enhanced CT scans, and that is a confounding factor that influences this subgroup analysis.

2. *“The model was built based on 2D CT slices (slice-by-slice) using a pretrained network that I believe is trained using natural images for object detection. Stage 1 (lung detection) and Stage 2 (nodule candidate detection) utilized 2D and 2.5D RGB input. There has been multiple advancement in medical image analysis in the 3D space such as the JOHOF package (<https://doi.org/10.1186/s41747-020-00173-2>) for lung lobe detection and nn-Unet (<https://doi.org/10.1038/s41592-020-01008-z>) for lesion segmentation that specifically designed for medical images and using NIFTI files as input. I would be more confident with the proposed method if the authors could show that their proposed model works on par (if not better) in comparison to the mentioned tools.”*

It is correct that all network components are in-fact 2D networks and have been pre-trained on natural images (i.e., the COCO dataset [2] for the lung and nodule candidate detection components, and ImageNet dataset [3] for the false positive reduction network). The false positive reduction network takes nine different cross sections from each nodule candidate (see Figure 2) and utilizes 3D information better than the other components. All technical details of the nodule detection pipeline are described in Supplementary Note 2.

In Supplementary Note 1, we validated the proposed architecture on the LUNA16 dataset with 10-fold cross validation. The model has a sensitivity of 96.1% at 1 FP/s and a competition performance metric (CPM) score of 94.5% (i.e., the sensitivity at 7 predefined false positive rates, namely 0.125, 0.25, 0.5, 1, 2, 4, and 8 FP/s). On this dataset, deep-learning based models from previous works obtained a sensitivity of 88.0%-93.6% at 1 FP/s and CPM scores of 77.5%-92.5% [4]. Hence, the proposed pulmonary nodule detection network can be regarded as state-of-the-art. We added the comparison of the models to Supplementary Note 1.

Regarding the reviewer's suggested 3D segmentation architectures, we had three main reasons for choosing an object detection network instead of a segmentation network. First, vanilla U-net segmentation models struggle with large foreground-background imbalances, which is the case for pulmonary nodules that can be a few voxels large. This can be addressed by taking small patches around the nodules before segmentation [5] or modifying the U-net architecture specifically for this task [6, 7]. Object detection networks share the same problem, but they generally work better for this problem thanks to the use of priors (e.g., anchors). Likely for this reason, segmentation networks are not as often used for pulmonary nodule detection as object detection networks [4]. Second, (one-stage) object detection models are computationally less expensive compared to segmentation models, because the decoder does not up-sample the coarse feature maps into a full-resolution segmentation map. Therefore, one-stage object detectors are popular in real-time applications and fast instance segmentation remains an active research field [8]. Third, bounding box labels are easier to acquire than segmentation labels. This was not important for the nodule detection task thanks to our segmentation software, but this reduced the efforts for developing the lung detector. We have elaborated on these choices in the nodule detection pipeline description in Supplementary Note 2.

The evaluation of the lung detection system can be found in Supplementary Note 4. The analysis shows that a small YOLOv5 model can accurately detect the lungs with single slices as input (i.e., precision and recall of 97% for slice-level detections) and this proved to be sufficient for our purposes. The use of 3D segmentation networks such as nn-Unet would be recommended when voxel-level precision labels are required (e.g., for obtaining lobe-level nodule location), albeit at the cost of higher computation requirements. We will further discuss the topic in our answer to review point 12 of Reviewer #2.

3. "Authors mentioned that semi-automated tool by reference [28] was utilized to delineate the tumour after radiologist identified the location of the lesion under interest. How did the authors validate the accuracy of the segmentation done by the tool afterwards? Since the output of the segmentation used as the ground truth, I assume some kind of quantitative measurements, or some visual corrections considered in that process (if so, author may want to consider to convey such information to the readers)."

We agree that this point needs further clarification in the manuscript. The radiologists both detected and segmented the pulmonary nodules and were able to directly make visual adjustments during this process. For the evaluation process, the segmentations were only used to estimate the equivalent nodule diameter and volume. Considering that a degree of interobserver variability can occur for nodule segmentation, we averaged the diameter and volume measurements from the different readers (see Table 2). We have clarified this process in the method section of the manuscript.

REVIEWER #2

1. *“This is a strong study, but the lack of prospective validation, the relatively small validation dataset which excluded 25% of scans for are substantial limitations.”*

We agree with the reviewer that these limitations need more explanation. In this answer, we will focus on explaining the reason for excluding the scans, while we will discuss our rationale for the retrospective study design and dataset size in review points 7 and 9. The higher exclusion rate can mainly be accounted for by the stratified sampling of patients with benign nodules, malignant nodules, or no (relevant) nodules. The largest exclusion group were patients who developed many pulmonary metastases (26/67, 39% of all exclusions). Patients with metastasized cancer can develop as many as hundred pulmonary metastases. It is impractical for our radiologists to annotate every lesion in the scan and therefore we set an upper limit of 15 lesions. Given that patients with metastasized cancers are relatively rare, the exclusion rate would be substantially lower when employing the model on the whole patient population. Nonetheless, we expect that the evaluation results generalize to patients with higher nodule counts than our threshold. Another important exclusion group were patients with extensive fibrosis or consolidations (e.g., due to severe interstitial diseases, haemorrhage, or pneumonia), accounting for 30% of all exclusions (20/67). These patients cannot be reliably screened for pulmonary nodules by both radiologists and AI systems, as their lungs contain many high attenuation areas that prevent correct localization and delineation of relevant nodules. The rationale of our exclusion criteria has been explained in the Eligibility criteria section in the Method section.

2. *“The recalibration of the model cutoff (Table 4) to compare against the radiologists is somewhat confusing and possibly not the best comparison. They recalibrated to a FP/s = 1, which is higher than any of the radiologists’ FP/s rate, so the sensitivities between AI and radiologists are difficult to compare. I think either recalibrating the AI to the FP/s rate of each radiologist would be a fairer comparison (essentially, comparing sensitivity at the same specificity), OR presenting the AUC (cutoff agnostic).”*

We appreciate the reviewer’s suggestion and realize that the purpose of this analysis could have been presented more clearly. The main purpose of the analysis is to put the detection results into perspective by comparing the stand-alone performance of the AI system with those of the radiologists. In clinical practice, the AI system would be fixed at an operating point, and it would be used consistently at this setting. Commercial AI systems typically operate at a FP rate of 1 or 2 FP/s on average. There is currently no consensus in the scientific community about the most optimal operating point for a pulmonary nodule detection system when used in a second reader or

concurrent reader setting. We chose to use an operating point of 1 FP/s and compare the stand-alone performance of the AI system at this operating point with the performance of our radiologists. This choice is based on previous research (see second paragraph in the Discussion section) and the opinion of our panel of radiologists.

The reviewer's feedback on this analysis is related to review point 4 about the claim of "expert-level" performance. The fairest comparison would be to evaluate all sensitivity scores at the same operating point (e.g., 1 FP/s), but this is not possible for the radiologists' performances due to the lack of confidence scores in the annotations. As a result, we cannot calculate an area under the Free-Response ROC curve [9], or the area under the ROC curve (AUROC) for patient-level analysis. An alternative analysis could be indeed to recalibrate the AI system for each comparison so that it matches the false positive rate of each reader. We argue that this is less representative of the real-world application of the AI system and therefore we have not included these results in our manuscript. However, as these results may still be interesting for the reviewer, we have added a table with the recalibrated results at the end of this document. These results show that the system reaches an equal or higher sensitivity for 14 out of 30 direct comparisons with a radiologist at the same false positive rate.

Based on the feedback of the reviewer, we have rewritten the manuscript to put less emphasis on the individual performances between the AI system and the radiologists and have put more emphasis on the possible increases in detection sensitivity when using an acceptable operating point of the AI system (see changes in red). We have decided to remove "expert-level" performance to prevent possible confusion. Hopefully these changes adequately address the feedback of the reviewer.

3. "Figure 5 is not the optimal way to present this data. A bidirectional bar chart should be used to show hazards/effect sizes/major differences in populations. A fairer comparison of the two populations would just use a standard bar chart, where the data have zero on the left and extend to the right, with different colours for the two groups. I would also like to see, rather than "nodules missed by the majority of radiologists," a marker showing the "most accurate, median, and least accurate radiologist" to better contextualize the AI results compared to the radiologist."

We thank the reviewer for the suggestion and have updated Figure 5 accordingly. The new figure shows a grouped, standard bar chart with different colours for the internal and external test set. New markers have been added that show the highest sensitivity, median, and lowest sensitivity of the radiologists for the given set of false negatives. Furthermore, small digits have been added that show the total number of false negatives or false positives per category to improve readability.

4. "Lastly, the main conclusion of the article is: "In conclusion, this study demonstrates that a DL-CAD system obtained an expert-level performance for the detection of benign actionable nodules, primary lung cancers, and pulmonary metastases in CT scans from a retrospective cohort of a routine clinical population." The study did not demonstrate expert-level performance, particularly for detection of primary lung cancer. The results show that the difference between expert and AI was not statistically significant ($p > 0.05$), but this does not prove expert-level performance, which would need to be evaluated with a non-inferiority or equivalency design. Specifically, the decrease in sensitivity (even if non-significant) for detecting early stage cancers is concerning because the end goal of nodule

detection is detection of cancers. Any missed benign nodule is actually an overall net positive to long term outcomes.”

As discussed in review point 2, we agree that the current method of analysis is suboptimal for determining “expert-level” discriminative power as this would indeed require a non-inferiority or equivalency design and FROC curves from the readers as well. The current FROC analysis and comparison with radiologists does however address the primary research question of validating an AI system in a non-screening setting and exploring its potential clinical value. To fully demonstrate the clinical utility of the proposed AI system, two sufficiently large groups of radiologists should read the scans with and without the help of AI assistance and then the results should be linked to actual clinical outcomes (e.g., preventing any delay in cancer diagnosis, improving patient survival, or reducing the number of follow-up CT studies). Although we agree that this is interesting, this is outside the scope of the presented manuscript.

Regarding the detection of primary lung cancers, the AI system missed 3 out of 59 malignant lesions (two in the internal test set and one in the external test set). As explained in the discussion section (fourth paragraph), two of these lesions were non-solid and one was juxtapleural. Non-solid pulmonary cancers are very rare and therefore not well represented in the training data (Mets et al. [10] report an incidence of 1.2% for non-solid nodules in a routine clinical setting, the incidence of malignant lesions will be lower). These lesions were not missed by the radiologists, so it is unlikely that these detection failures would lead to a decrease in sensitivity in a concurrent reading setting. For the development of autonomous applications, special attention should be given to the detection of these rare lung cancers.

The end goal is indeed the detection of cancers, but we think that an AI system should be able to detect all clinically relevant lesions (≥ 5 mm, not perifissural or calcified) in routine clinical practice in regardless of their malignancy status. In this way, the AI system can support radiologists in their decision-making process in accordance with nodule management guidelines such as Fleischner [11] or BTS [12]. Furthermore, any missed actionable benign nodule by the AI system can hurt radiologists’ confidence in the system, especially when the nodule has malignant features (i.e., spiculation).

5. *“There are a few typos throughout the manuscript.”*

We have carefully reread the manuscript and corrected any typos that we could find (see changes in red).

6. *“The emphasis on incidentally identified nodules is very useful, but the main manuscript should provide a description of the total pool of chest CTs the external validation dataset was sampled from. How many scans (coronary scans, PE studies, contrast VS not...) would have been excluded from analysis because they do not meet technical eligibility for radiomic analysis?”*

The total pool of chest CT studies and exclusions per institution have been described in Figure 1. In hospital A, 14,943 adult patients underwent 34,689 thorax and thorax-abdomen CT scans in the period of 2018-2020. In hospital B, 12,739 patients underwent 22,621 thorax and thorax-abdomen CT scans in the same period. All types of CT scans were included (including CTA) to ensure generalizability to the clinic, only CT scans with thick slices (> 3 mm), missing slices (i.e., data

corruption), or very low volume (< 50 slices) were excluded. Based on these criteria, 323 studies (0.9%) were excluded at hospital A and 33 studies (0.15%) were excluded at hospital B. To further clarify this process, we have added a description to the test datasets characteristics section in the results section.

7. "The balanced datasets with 4 categories are useful for validation purposes but introduce another problem of generalizability which call for prospective validation."

We understand the point of the reviewer and therefore we added a recommendation in the Discussion section to evaluate the model in a prospective clinical setting. A retrospective study design with a balanced dataset was the most feasible option given the low prevalence of malignant pulmonary nodules in a non-screening population [13]. For a prospective validation study, one would need a relatively long study period (at least a couple of years) to ensure a sufficiently large group of patients with early-stage lung cancer or seek collaboration with more than two institutions.

8. "Geographic differences in nodule prevalence and etiology are likely to substantially impact the external validity of the model as well."

We agree that nodule prevalence and etiology can influence the generalizability of the AI system. For this reason, we evenly sampled data from different nodule categories (benign, metastases, and primary cancers) and conducted a separate analysis per category to minimize the effect of these factors. The prevalence of lung diseases (e.g., ILD, pneumonia, etc.) may also affect the external validity of the research findings, although these conditions were included in our dataset whenever nodule assessment was still possible. We believe that geographic differences in nodule prevalence and etiology will indeed become important in the evaluation of the clinical impact of the proposed AI system in a prospective setting. The proposed AI system will be made freely accessible for scientific uses on the public platform Grand-Challenge, where other researchers can validate the system performance on larger, geographically different, cohorts.

9. "200 scans were selected for validation to achieve sufficient statistical power but the power calculation and assumptions used are not explicitly described."

We have not performed a power calculation and realize that the phrase "to achieve sufficient statistical power" can imply this calculation. We selected 200 scans, 100 per hospital, to obtain relevant results. We would have liked to make the test datasets larger, but the annotation process was costly due to our chosen reference standard. Each of the five thoracic radiologists had to annotate all 200 CT scans and the annotation time per radiologist was 12 hours on average (this task was done on a voluntary basis). The malignancy status verification of the nodules (done by two additional radiologists), preliminary reading, and postprocessing of the data took many more hours. In our opinion, the large panel of radiologists was essential to address the large interobserver variability in nodule detection [14]. We have updated the explanation of our dataset size in the "Method and materials" section (see changes in red).

10. "Test dataset should not be under the results section but in the material and methods section since it explains the data design process."

We explain the data design process in the “Datasets” and “Eligibility criteria” sections in the Materials and Methods section, but we would like to report the flow of participants in the Results section. While we understand the rationale of moving the “Test datasets characteristics” section, we decided to adhere to the reporting style of the STARD 2015 guidelines [15] that is commonly used in diagnostic accuracy studies. Hopefully the reviewer agrees with our point of view.

11. “Figures 3 and 4 can be combined as they showcase the same quantitative result.”

We appreciate the suggestion of the reviewer, but the FROC curves in Figures 3 and 4 are calculated in a different way. Figure 3 shows a FROC curve based on the full reference standard of five radiologists, whereas Figure 4 shows an averaged FROC curve of five different comparisons with a slightly different reference standard (see Method section, section Analysis). Although the figures are indeed similar, we think it is best to present them separately to avoid misinterpretation.

12. “In the supplementary material, the authors claim that the object detection approach is far superior compared to segmentation approaches. There is no evidence to prove this either through citations or previous methods that have proved to be slower than the employed method in the paper.”

We believe the reviewer refers to the explanation of our model choice for the task of lung detection in Supplementary Note 2. We agree with the reviewer that this aspect needs more explanation. As explained in our answer to the second question of reviewer #1, most segmentation models, such as U-Net, have a decoder part for upsampling the features maps to recover spatial details for precisely delineate object boundaries. Hence, object segmentation is generally more precise, but also more computationally expensive than object detection (especially when using 3D convolutions).

In that regard, thresholding-based segmentation approaches (combined with morphological operations) do not have this disadvantage. These approaches are very fast and have been often used for lung segmentation tasks in a nodule detection pipeline [16-18]. However, this method only performs well in case of absent or minimal presence of lung pathologies [19]. In a routine clinical setting, pathological conditions are usually present (e.g., consolidations, pleural effusions, fibrosis, etc.) and therefore these traditional methods can be fragile. We have updated our description of the nodule detection pipeline and added references to substantiate our design choices.

13. “The need for a lightweight and heavy weight YOLOv5 model in the lung detection pipeline and the nodule detection pipeline is vague and needs further explanation.”

We agree that this design choice needs further explanation. The pipeline configuration has been determined empirically: for each task, we started with the lightest model configuration and increased the model capacity until no performance gain could be achieved. For the lung detection component, it turned out that the lightest model was already sufficient and that there was no need for multi-slice input images (unlike the nodule detection component). The nodule detection component needed more capacity to fully utilize the multi-slice input images (due to higher dimensional data). We have added this information to the nodule detection pipeline description in Supplementary Note 2.

We thank you and the reviewers again for their conscientious revisions and comments, and we hope to have adequately addressed all points of feedback.

REFERENCES

1. Purysko, C. P., Renapurkar, R. & Bolen, M. A. When does chest CT require contrast enhancement? *Cleve. Clin. J. Med.* **83**, 423–426 (2016).
2. Lin, T.-Y. et al. Microsoft COCO: Common objects in context. Preprint at <https://arxiv.org/abs/1405.0312> (2014).
3. Deng, J. et al. ImageNet: A large-scale hierarchical image database. in *2009 IEEE Conference on Computer Vision and Pattern Recognition* 248–255 (IEEE, 2009).
4. Gu, Y. et al. A survey of computer-aided diagnosis of lung nodules from CT scans using deep learning. *Comput. Biol. Med.* **137**, 104806 (2021).
5. Gu, D., Liu, G. & Xue, Z. On the performance of lung nodule detection, segmentation and classification. *Comput. Med. Imaging Graph.* **89**, 101886 (2021).
6. Keetha, N. V., P, S. A. B. & Annavarapu, C. S. R. U-Det: A Modified U-Net architecture with bidirectional feature network for lung nodule segmentation. *arXiv [eess.IV]* (2020).
7. Chiu, T.-W., Tsai, Y.-L. & Su, S.-F. Automatic detect lung node with deep learning in segmentation and imbalance data labeling. *Sci. Rep.* **11**, 11174 (2021).
8. Bolya, D., Zhou, C., Xiao, F. & Lee, Y. J. YOLACT: Real-time Instance Segmentation. *arXiv [cs.CV]* (2019).
9. Bandos, A. I., Rockette, H. E., Song, T. & Gur, D. Area under the free-response ROC curve (FROC) and a related summary index. *Biometrics* **65**, 247–256 (2009).
10. Mets, O. M. et al. Subsolid pulmonary nodule morphology and associated patient characteristics in a routine clinical population. *Eur. Radiol.* **27**, 689–696 (2017).
11. MacMahon, H. et al. Guidelines for management of incidental pulmonary nodules detected on CT images: From the Fleischner Society 2017. *Radiology* **284**, 228–243 (2017).
12. Callister, M. E. J. et al. British Thoracic Society guidelines for the investigation and management of pulmonary nodules. *Thorax* **70 Suppl 2**, ii1–ii54 (2015).
13. Hendrix, W. et al. Trends in the incidence of pulmonary nodules in chest computed tomography: 10-year results from two Dutch hospitals. *Eur. Radiol.* (2023) doi:10.1007/s00330-023-09826-3.
14. Pinsky, P. F., Gierada, D. S., Nath, P. H., Kazerooni, E. & Amorosa, J. National lung screening trial: variability in nodule detection rates in chest CT studies. *Radiology* **268**, 865–873 (2013).
15. Cohen, J. F. et al. STARD 2015 guidelines for reporting diagnostic accuracy studies: explanation and elaboration. *BMJ Open* **6**, e012799 (2016).
16. Liao, F., Liang, M., Li, Z., Hu, X. & Song, S. Evaluate the malignancy of pulmonary nodules using the 3-D deep leaky noisy-OR network. *IEEE Trans. Neural Netw. Learn. Syst.* **30**, 3484–3495 (2019).
17. Han, Y. et al. Pulmonary nodules detection assistant platform: An effective computer aided system for early pulmonary nodules detection in physical examination. *Comput. Methods Programs Biomed.* **217**, 106680 (2022).

18. Zheng, S. *et al.* Automatic pulmonary nodule detection in CT scans using convolutional neural networks based on maximum intensity projection. *IEEE Trans. Med. Imaging* **39**, 797–805 (2020).
19. Mansoor, A. *et al.* Segmentation and image analysis of abnormal lungs at CT: Current approaches, challenges, and future trends. *Radiographics* **35**, 1056–1076 (2015).

Table 1. Comparison between the detection performance of the AI model and individual readers for benign actionable nodules, primary cancers, and pulmonary metastases in the internal (hospital A) and external (hospital B) test set with matched false positive rates.

	FP/scan	Actionable benign nodules		Primary cancers		Metastases	
		Sensitivity (%)	p	Sensitivity (%)	p	Sensitivity (%)	p
Internal							
Radiologist 1	0.7 (0.5, 0.9)	98 (94, 100)	.06	100 (100, 100)	.46	77 (66, 88)	.72
Recalibrated AI	0.6 (0.4, 0.9)	89 (79, 96)	.06	93 (82, 100)	.46	79 (72, 86)	.72
Radiologist 2	0.1 (0.0, 0.2)	79 (69, 89)	.53	100 (100, 100)	.02	32 (18, 47)	<.001
Recalibrated AI	0.1 (0.0, 0.1)	72 (59, 84)	.53	78 (60, 92)	.02	48 (36, 61)	<.001
Radiologist 3	0.2 (0.1, 0.3)	89 (81, 97)	.37	96 (87, 100)	.22	70 (57, 83)	.11
Recalibrated AI	0.2 (0.1, 0.3)	82 (71, 92)	.37	81 (66, 95)	.22	63 (54, 74)	.11
Radiologist 4	0.5 (0.3, 0.7)	79 (70, 88)	.36	100 (100, 100)	.25	45 (37, 54)	<.001
Recalibrated AI	0.5 (0.3, 0.7)	88 (77, 96)	.36	89 (75, 100)	.25	76 (67, 85)	<.001
Radiologist 5	0.8 (0.5, 1.0)	77 (59, 95)	.09	100 (100, 100)	.52	87 (80, 93)	.01
Recalibrated AI	0.7 (0.5, 1.1)	89 (79, 97)	.09	93 (82, 100)	.52	79 (72, 86)	.01
External							
Radiologist 1	0.7 (0.4, 1.0)	90 (81, 96)	.32	97 (90, 100)	>.99	81 (74, 91)	.16
Recalibrated AI	0.7 (0.4, 1.0)	96 (89, 100)	.32	94 (83, 100)	>.99	88 (83, 93)	.16
Radiologist 2	0.1 (0.1, 0.2)	70 (60, 78)	.66	91 (78, 100)	>.99	66 (59, 73)	.08
Recalibrated AI	0.1 (0.1, 0.2)	74 (59, 85)	.66	88 (76, 100)	>.99	76 (68, 86)	.08
Radiologist 3	0.3 (0.1, 0.4)	94 (86, 99)	.04	94 (86, 100)	>.99	75 (65, 87)	.53
Recalibrated AI	0.3 (0.1, 0.4)	83 (73, 92)	.04	91 (80, 100)	>.99	79 (71, 88)	.53
Radiologist 4	0.5 (0.4, 0.7)	66 (53, 78)	<.001	97 (90, 100)	>.99	61 (50, 74)	<.001
Recalibrated AI	0.5 (0.3, 0.7)	91 (81, 97)	<.001	94 (83, 100)	>.99	87 (81, 92)	<.001
Radiologist 5	0.6 (0.4, 0.7)	86 (75, 95)	.16	97 (90, 100)	>.99	88 (83, 93)	>.99
Recalibrated AI	0.5 (0.4, 0.8)	94 (87, 99)	.16	94 (83, 100)	>.99	88 (83, 93)	>.99

Note. The AI-system was recalibrated for each comparison to match the FP/s of the reader. Note that the exclusion list and reference standard also slightly varied per reader (latter only for benign nodules).

Abbreviations: FP/scan = average number of false positives per scan.

REVIEWERS' COMMENTS:

Reviewer #1 (Remarks to the Author):

The authors have addressed the concerns sufficiently. Thank you for further investigating point 1, good to know that the proposed method works in both scan types. Acknowledge the reasons for not choosing 3D algorithms in point 2.

I have no other comments thus recommending the work for publication.

Reviewer #2 (Remarks to the Author):

Thank you for your revisions, great work.